# Self-Reported Oral Health Related Behaviour and Gum Bleeding of Adolescents in Slovakia in Relation to Socioeconomic Status of Their Parents: Cross-Sectional Study Based on Representative Data Collection

**DOI:** 10.3390/ijerph16142484

**Published:** 2019-07-12

**Authors:** Silvia Timková, Peter Kolarčik, Andrea Madarasová Gecková

**Affiliations:** 11st Dental Clinic, Faculty of Medicine, P.J. Šafárik University in Košice and University Hospital of Luis Pasteur, Trieda SNP 1, 04011 Košice, Slovakia; 2Department of Health Psychology, Faculty of Medicine, P.J. Šafárik University in Košice, Trieda SNP 1, 04011 Košice, Slovakia; 3Olomouc University Social Health Institute (OUSHI), Palacký University in Olomouc, Univerzitní 22, 77111 Olomouc, Czech Republic

**Keywords:** oral health related behaviours, oral hygiene, gum bleeding adolescence, socioeconomic status, parents’ education, family affluence scale

## Abstract

Background: Oral health strongly affects overall health and is related to many factors. The aim of our study was to analyse oral health related behaviours (OHRBs) and gum bleeding among Slovak adolescents and assess the effect of socioeconomic factors on the outcomes. Methods: Data from the Health Behaviour in School-aged Children study (HBSC) were used (N = 8896, age range = 10–16 years, M = 13.4; SD = 1.4; 50.9% boys). Sociodemographic and socioeconomic indicators and frequency of OHRBs (dental hygiene, toothbrush changing, preventive check-up) and gum bleeding were collected. Effects of sociodemographic and socioeconomic variables on outcome variables were analysed by binary logistic regression. Results: We found that prevalence of OHRBs slightly decreases with age, and worse outcomes were reported by boys compared to girls (OHRB odds ratio range 0.45–0.75, (95% C.I. range 0.40–0.91), gum bleeding 1.38 (95% C.I. 1.19–1.61), *p* < 0.05). OHRBs were in most cases significantly associated with socioeconomic variables, lower affluence predicts worse outcomes (odds ratio range 0.76–0.88 (95% C.I. range 0.68–0.96), *p* < 0.05). Conclusion: Our study provides representative findings on ORHBs in Slovakia and shows important associations of socioeconomic factors related to adolescents’ oral health issues.

## 1. Introduction

The presence of dental caries, gingivitis, periodontitis and the overall level of oral health strongly affects the overall health of children and adolescents in their future [1,2,3,4,5,6,7,8]. The presence of gingivitis or periodontitis may induce a chronic inflammatory response in the body [3,5,9,10]. Chronic periodontitis is a potential risk factor for atherosclerosis and other cardiovascular diseases, which can occur in adults [10,11,12]. Gingivitis or early signs of problems with gums manifest as bleeding from gums during tooth brushing that may be easily screened by questionnaires [13].

Likewise, the overall health of an individual, positive or negative, can affect also the oral health of an individual. Insufficient oral health leads to a decrease in quality of life of the individual. The basic prevention against oral health complication is regular oral hygiene, tooth brushing is recommended twice a day [14] and regular dental preventive check-ups at least once, but more preferably twice, a year for children and adolescents [15].

At the present, dental care in Slovakia is provided in 2428 dental clinics by 3198 dentists. One third of all dental clinics are situated in Bratislava and Košice, the two biggest cities in Slovakia. So the availability of dental care is not problematic in big cities, however it is a problem in particular regions in Slovakia. Also, we do not have enough dental clinics with dentists with a paediatric specialization, which makes proper dental and oral health care among children less available. Only 76% of children and adolescents had preventive examinations once a year [16], but it is recommended to have preventive examinations twice a year for children until the age of 18. The need for treatment in children and adolescents in this age group was 57.71% [16]. We find this proportion of the need for treatment in the paediatric age group as a very high compared to other developed countries and the causal factors of this still need to be identified.

Generally, index DMF (so called decay-missing-filled index) is used as an indicator of oral health and it quantifies the presence of decayed (D), filled (F) and missing teeth (M) [17]. The higher the DMF index of an individual the worse his or her oral health.

In Slovakia, in 2017, the DMF index of oral health in 12 year olds was 1.71 and the DMF index of oral health in 15 year olds was 2.78, which highlights the problematic oral health among Slovak adolescents [16]. The WHO Health Program aims to achieve a DMF level of 1.5 among children as their goal by 2020 [18,19]. The DMF index in Slovakia is slowly decreasing, which means that oral health is slowly improving, but still lagging behind the desired European level [18].

Relatively large geographical differences were reported in the oral health among 12 and 15 year old children between the regions of Slovakia [16]. It appears that there are areas with better oral health, such as the Bratislava and Nitra regions, but also regions with noticeably worse oral health, especially in the regions of Žilina, Banská Bystrica and Prešov regions. The average DMF index for 12 and 15 year old children (DMF 1.14 and 1.76, respectively) in the Bratislava region is lower comparing it to the DMF in the Žilina region (DMF 2.10 and 3.55, respectively) or Košice region (DMF 1.68 and 2.70, respectively) [16]. These high regional disparities need to be understood and eliminated, but no effective effort has been made towards this goal yet.

This is demonstrated in the main focus of oral health monitoring in Slovakia, where only the two standard factors of oral health have been monitored for years, specifically: the level of oral hygiene of an individual and the intake of sugar and sweetened soft-drinks. Other risk factors have been omitted from monitoring and the lack of understanding of the contributing factors of oral health deterioration is what provided only a limited understanding of the problem and the ineffective interventions resulting from it. Obviously, socioeconomic status is a strong factor affecting oral health although no relevant research has studied this association in a Slovak setting. Several studies have highlighted this link, where it is assumed that oral health is linked to the parameters of parental education, their income, gender, as well as their living conditions and other parameters [19,20,21,22,23,24,25,26,27]. It might be expected that better oral health related behaviours and prevalence of healthy gums would be associated with higher socioeconomic status of the parents, such as higher education level, better family affluence.

The aim of our study was to analyse self-reported oral health related behaviours and gum bleeding among Slovak adolescents and assess the effect of socioeconomic factors on the prevalence of the selected oral health related indicators.

## 2. Materials and Methods 

### 2.1. Sample and Procedure

We used data from the Health Behaviour in School-aged Children (HBSC) study conducted in 2018 in Slovakia. HBSC used a two-step sampling to obtain a representative sample. In the first step, 140 larger and smaller elementary schools located in rural and urban areas from all regions of Slovakia were asked to participate. These were selected using weighted random selection (type of school and region) from a list of all eligible schools in Slovakia (N = 1616) obtained from the Slovak Institute of Information and Prognosis for Education. School response rate (RR) was 73.6%. In the second step, we obtained data from 8902 adolescents from the fifth to ninth grades of elementary schools and prima to quarta of eight year grammar schools in Slovakia. Eligible respondents were all pupils in selected schools with an age range from 10 to 17 years old (mean age 13.4; SD = 1.4; 50.9% boys). We excluded pupils from participating in the data collection who provided their parent’s disagreement with participation in the HBSC study, pupils who were not present in the school and pupils who individually refused to participate. For the purposes of the study we excluded also 17 year olds who comprised small group (N = 8).

The study was approved by the Ethics Committee of the Faculty of Medicine at the P.J. Šafárik University in Kosice (date of approval: 1.12.2017). Parents were informed about the study via the school administration and could opt out if they disagreed with their child’s participation. Participation in the study was voluntary and anonymous with no explicit incentives provided for participation.

### 2.2. Measures

Data for the present analyses were collected using questionnaires from the standardized research protocols for the 2017/2018 WHO-collaborative HBSC study. We collected data on sociodemographic characteristics (age, gender), socioeconomic indicators (family affluence scale (FAS), score from 0–13, higher score indicates more affluence) [28], self-rated family well-off (How well-off do you think your family is?, answer categories: 1 =very well-off to 5—not well-off at all), parents’ employment status (Does your father/mother have a job?, answer categories: 1—yes, 0—no) and education level (What school, what kind of education has your father/mother finished?, answer categories: 1—elementary, 2—apprenticeship, 3—secondary, 4—university) and data on oral health promoting behaviour and oral health indicators from following questions:

Dental hygiene—“How often do you brush your teeth?” (Answer categories: 1 = More than once per day, 2 = Once per day, 3 = More than once per week, but not daily, 4 = Less than once per week, 5 = Never). The answer categories were dichotomized as 1 = twice a day vs. 0 = once a day and less often.

Toothbrush changing—“How often do you change your toothbrush?” (Answer categories: 1 = At least once per month, 2 = At least once every 3 months, 3 = At least once every 6 months, 4 = At least once per year). The answer categories were dichotomized as 1 = at least once every 3 months and more often vs. 0 = less often.

Preventive dental check-up—“Did you visit the dentist last year?” (Answer categories: 0 = no, 1 = yes). 

Gum bleeding—“Do your gums bleed while you are brushing your teeth?” (Answer categories: 1 = Always, 2 = Often, 3 = Occasionally, 4 = Sometimes, 5 = Never). The answer categories were trichotomized as 2 = always and often, 1 = occasionally and sometimes, 0 = never.

### 2.3. Data Analysis

As a first step, we recategorized outcome variables and calculated descriptive statics to provide outcome prevalences among the respondents and an overview of sociodemographic and socioeconomic variables. As question non-response occurred in non-systematic way, our total responses on particular tested variables varied. For data analysis we used list-wise case deletion regarding each analysis. Differences between boys and girls were tested using Pearson’s chi-squared test for categorical variables and the Mann–Whitney *U*-test for continual variables. In the second step, we tested crude effects of sociodemographic and socioeconomic variables on outcome variables in binary logistic regression and mutually adjusted effects of all predicting variables in one combined model. As a final step, we used the forward conditional variable selection method in binary logistic to establish a predictive model with significant predictors for each oral health outcome from all potential predictors used in this study. The forward conditional variable selection method is a stepwise selection method with variable entry testing based on the significance of the score statistic, and removal testing based on the probability of a likelihood-ratio statistic based on conditional parameter estimates (ref. IBM website). All statistical analyses were performed using statistical software package IBM SPSS 22.0.

## 3. Results

Sociodemographic and socioeconomic composition of the sample is presented in Table 1. There were no statistically significant differences in the employment status of both parents and in family affluence scale categories or FAS total score between boys and girls. We can also see trivial but statistically significant differences in age and family affluence reported by boys and girls. Boys reported slightly better family affluence. The number of respondents varied regarding the oral health outcomes. The dental hygiene question was asked across all age categories (N = 8673), but the rest of the oral health questions were only given to children aged 13 and older (valid responses varied from N = 5067 to N = 5093).

Regular tooth brushing at least twice a day was reported by 63% of the respondents, with a higher proportion among girls compared with boys (Table 2). This proportion slightly decreases with older age categories. Regularly changing toothbrushes, at least once every three months, was reported by 78% of respondents aged 13 and above, with a similar proportion among boys and girls, without a statistically significant difference. Visiting the dentist for a preventive check-up at least once in the last 12 months was reported by 84% respondents also with no statistically significant difference between boys and girls. The proxy indicator of oral health, no bleeding of gums while brushing teeth, was reported by almost 45% of respondents. Always or often bleeding of gums was reported by almost 7% of respondents (Table 3).

Analyses of the crude effects of individual sociodemographic and socioeconomic variables on prevalence of the outcome variables showed that better dental hygiene (tooth brushing) is associated with lower age, being a girl, coming from a family that is more affluent, having parents with a higher education level and having employed parents. Changing toothbrushes at least once every three months is not associated with age, but it is associated with being a girl, coming from a more affluent family and having parents with a higher education level. Employment status of the parents was not associated with the frequency of toothbrush changing. Attending dental preventive check-ups is associated with the same predictors as dental hygiene, except with the opposite association of gender, where being a boy predicts attendance of check-ups. A higher chance to not report any gum bleeding is associated with an older age, being a girl, a more affluent family, but with parents with a lower education level. Employment status of the parents was not related to gum bleeding (Table 4).

Assessment of individual variables and their crude effects on the outcomes might be misleading and could provide invalid conclusions. Therefore, we tried to also provide a model where all potential explaining variables are mutually adjusted and only the strongest predictors remained (Table 5). Thus, we could see how those explaining variables affect the effect of each other and provide results that are more realistic. After these analyses, we could see that age became a statistically insignificant predictor in most outcomes except gum bleeding. Family affluence score, the mother’s education and both parents’ employment status are not relevant to consider in explaining the frequency of changing toothbrushes and gum bleeding. The father’s education is not significantly associated with gum bleeding.

The most complex predictive model was established for dental hygiene (tooth brushing). Reporting of brushing teeth at least twice a day is associated with being a girl, higher family affluence and better subjective affluence, having a mother and father with university level education and having an employed mother. Regularly changing toothbrushes at least once every three months is also associated with being a girl, better subjective affluence and having a father with university level education, but not with the mother’s education level. Attendance of preventive dental check-ups is associated with the same predictors as dental hygiene with the addition of the father being employed. Absence of reported gum bleeding is also associated with being a girl, better affluence and older age.

Using the forward conditional method for eliminating insignificant outcome predictors echoed the results from the previous combined model. The only difference is the involvement of the father having a high education level as a statistically significant predictor of no bleeding of the gums compared to the previous combined model (Table 6).

## 4. Discussion

The aim of the presented study was to provide representative data on selected self-reported oral health indicators and attendance of preventive dental check-ups of Slovak adolescents and to provide some insights on the effect of sociodemographic variables on the prevalence of those oral health indicators. We found that most of our respondents reported brushing their teeth twice a day, changing toothbrushes at least once every three months and had a dental preventive check-up in last 12 months, as is generally recommended. Gum bleeding while brushing teeth was reported by a relatively large proportion of the adolescents, which indicates problems with oral health and might be indicator of future oral health complications. We also found that oral health related behaviours and gum bleeding are significantly associated with socioeconomic variables, respondents from families that are more affluent or reporting better affluence and having university educated father have higher chances of reporting better oral health outcomes. Gender was also significantly associated with oral health outcomes, showing girls reporting better oral health outcomes.

The proportion of adolescents reporting tooth brushing twice a day in Slovakia is lower compared with other European countries involved in HBSC studies and in all three age groups falls in the second half of the country rankings with the lowest rank in 15 year olds [29]. But similarly to other studies, regular tooth brushing is lower among older adolescents and among boys [29,30]. Similarly, attending a dental preventive check-up at least once a year was lower compared to a study conducted in the US [31] and it has a similar tendency to decrease with age. According to the Australian Health Policy Collaboration [32], dental check-up frequency should be related to an individual’s risk for developing oral disease as well as to detect or review any signs and symptoms. Early detection of disease allows for appropriate and timely intervention by dental professionals. Furthermore, regular check-ups allow for dental professionals to provide on-going preventive advice on lifestyle risk factors, such as dietary advice as good education and motivation for at-home oral hygiene practices which are essential for primary prevention of oral diseases.

In Slovakia, the DMF epidemiological index, which detects hard tooth damage in children and adolescents, is much higher than the WHO targets set and getting worse from age 12 to 15. It is necessary to determine the various related factors which could improve the overall value of the DMF index and thereby improve the health of the patients. From the results of the survey conducted, the interest in children and adolescents should be significantly higher if we want to improve the quality of their oral health.

In the first place, it is necessary to educate children and adolescents, as well as their parents, that teeth should be cleaned twice daily. Cleaning teeth only once daily leads to an increased incidence of caries, consequently to an increased need for teeth treatments, or extensive damage of teeth leading to tooth loss itself. Similarly, tooth brushing only once a day results in a higher risk of developing gum inflammation that aggravates with age and is a precursor to the development of a severe inflammatory disease of the periodontitis that strongly affects the overall health of individuals. Furthermore, it is necessary to explain that a toothbrush needs to be changed every two months because of its wear which reduces its effectiveness and because the brush itself may be the source of the accumulation of pathogenic bacteria that cause gum disease.

Epidemiologic data on periodontal diseases are of very poor quality and are absent from several European Member states [33]. We also did not find relevant sources for the comparison of the prevalence of gum bleeding among children or adolescents and regularly changing toothbrushes. In general, it has been suggested that over 50% of the European population suffer from some form of periodontitis and over 10% have severe diseases [33]. From a clinical point of view, we can say that the prevalence of gum bleeding reported in our study is not satisfactory and indicates potential oral health issues in the future. Regular changing of toothbrushes was relatively acceptable in the sample although it was dependent on family affluence. Buying a toothbrush might be a considerable expense for lower affluent families and so they might try to postpone changing the toothbrush as long as possible. Or such families just simply do not place as much importance on oral health as more affluent families might.

Socioeconomic differences found in our study echo findings of number of other studies dealing oral health [19,21,22,25,26,27,34]. It was proven that worse socioeconomic position is associated with worse oral health regarding clinical and self-reported outcomes [35].

Despite that, socioeconomic factors are not fully implemented in oral health and dental health care intervention. Socioeconomic factors should be considered as important risk factors contributing to oral health inequalities. For example, patients from a lower socioeconomic status environment should be targeted primarily and dental check-ups twice a year should be actively requested/demanded by dentists because there is a higher potential risk of dental carries and other oral health complications due to the lower teeth brushing frequency and higher gum bleeding. It is worthwhile to emphasize the need to systematically increase visits for preventive examinations in children and adolescents to twice annually. It is insufficient to a provide preventive dental check-up for children only once a year. Also, it is a bad sign that attendance of preventive dental check-ups is reducing with increasing age of children and adolescents. Also, the percentage of children with gum bleeding is very high. The preventive check-ups are the specific place where children and their parents need to be constantly educated. Improving the well-being of children and adolescents can only be done through continual and patient education.

### Strengths and Limitations

We consider the fact that this is the first study assessing the association of socioeconomic factors on self-reported oral health related behaviour and gum bleeding among Slovak adolescents, which was lacking for a long time in the dental community, to be a strength of our study. Another strength is the large and nationally representative sample based on internationally elaborated study protocol.

The limitations of the present study should be noted as well. First, the cross-sectional design allows us to only make inferences about associations and not about causality. Second, the self-reported nature of the data did not allow us to get an actual picture of the children’s oral health status which might only be possible using a dental examination, which was not feasible within the study design. However, the items used for monitoring the epidemiological situation in Slovakia were specifically selected to cover the main issues dealt with in oral health promotion and problem prevention. Third, we did not study the effect of potential confounders that might affect the presented associations to a certain degree. However, we aimed to provide a clear message on the associations of socioeconomic factors and the relevance of those factors for individual outcomes. Assessing various confounders was not the aim of our study. The fourth study limitation is the relatively high number of missing responses to individual questions covered by this study which accumulate in the regression models and lower the sample size. Regarding the nature of the study we consider those nonresponses as random and not affecting the associations reported here but lowering the statistical power of the sample. The last limitation is using different lengths of the questionnaire in three age groups (10–12, 13–14, 15 and over) because of the different ability to fill in the questionnaire in those age groups. The youngest respondents had the shortest questionnaire and thus the questions on toothbrush changing, preventive dental check-ups and gum bleeding were not included in their questionnaire. Within such a complex study as HBSC, it is not always feasible to cover all topics in all age categories and some selection had to be made in the favour of effectivity and feasibility.

## 5. Conclusions

Our study provides representative findings on oral health related behaviour and gum bleeding in Slovakia, showing that prevalences of tooth brushing and dental check-ups are below the average of European countries and with a relatively high proportion of tooth brushing among adolescents aged 10–16 and of dental check-ups and gum bleeding among adolescents aged 13–16. We found that girls tended to report better oral health outcomes compared to boys and that younger respondents tended to report better oral health outcomes compared to older ones. The study also supports the importance of socioeconomic factors and their association with oral health outcomes, indicating that the lower the socioeconomic situation of an adolescent’s family, the worse the adolescent’s health is.

## Figures and Tables

**Table 1 ijerph-16-02484-t001:** Description of sociodemographic and socioeconomic variables of the respondents.

		Whole Sample		Boys		Girls		Difference Between Boys and Girls
		N	%	N	%	N	%	
*Categorical variables*								
**Gender**								
	Boys	4529	50.9	- - -	- - -	- - -	- - -	
	Girls	4365	49.1	- - -	- - -	- - -	- - -	
	Total	8894	100.0	- - -	- - -	- - -	- - -	
**Father employment**								
	unemployed	470	6.6	235	6.6	235	6.5	
	employed	6671	93.4	3305	93.4	3366	93.5	0.848
	Total	7141	100.0	3540	100.0	3601	100.0	
**Mother employment**								
	unemployed	886	12.4	425	12.0	461	12.8	
	employed	6258	87.6	3108	88.0	3150	87.2	0.345
	Total	7144	100.0	3533	100.0	3611	100.0	
**Family affluence (FAS) categories**								
	low	2035	30.7	964	29.7	1071	31.6	
	middle	1980	29.9	965	29.7	1015	30.0	0.129
	high	2616	39.5	1317	40.6	1299	38.4	
	Total	6631	100.0	3246	100.0	3385	100.0	
**Family affluence categories**								
	very well-off	2022	29.5	1075	31.8	947	27.2	
	quite well-off	3016	43.9	1462	43.2	1554	44.7	
	average	1595	23.2	741	21.9	854	24.5	*p* < 0.001
	not very well-off	182	2.7	73	2.2	109	3.1	
	not at all well-off	48	0.7	32	0.9	16	0.5	
	Total	6863	100.0	3383	100.0	3480	100.0	
*Continual variables*								
**Age**								
	Mean (SD)	13.37 (1.44)		13.43 (1.45)		13.31 (1.43)		
								*p* < 0.001
	Total (N)	8894		4529		4365		
**Family affluence** (continual)								
	Mean (SD)	2.01 (0.83)		1.97 (0.84)		2.05 (0.83)		
								*p* < 0.001
	Total (N)	6863		3383		3480		
**Family affluence (FAS)**								
	Mean (SD)	7.74 (2.48)		7.80 (2.45)		7.69 (2.52)		0.056
	Total (N)	6631		3246		3385		

**Table 2 ijerph-16-02484-t002:** Proportion of respondents brushing their teeth twice a day stratified according to age categories (13–16 years of age) and gender.

		10 Years Old		11 Years Old		12 Years Old		13 Years Old		14 Years Old		15 Years Old		16 Years Old		All Age Categories	
		n	%	n	%	n	%	n	%	n	%	n	%	n	%	n	%
**Whole sample**																	
	once a day and less	115	32.3	564	36.2	630	36.6	673	36.6	680	37.2	483	38.8	55	46.2	3200	36.9
	twice a day	241	67.7	994	63.8	1093	63.4	1165	63.4	1146	62.8	762	61.2	64	53.8	5465	63.1
	Total	356	100.0	1558	100.0	1723	100.0	1838	100.0	1826	100.0	1245	100.0	119	100.0	8665	100.0
**Boys**																	
	once a day and less	58	36.0	330	43.8	375	42.7	437	45.9	423	46.3	332	49.6	38	49.4	1993	45.2
	twice a day	103	64.0	424	56.2	503	57.3	516	54.1	491	53.7	337	50.4	39	50.6	2413	54.8
	Total	161	100.0	754	100.0	878	100.0	953	100.0	914	100.0	669	100.0	77	100.0	4406	100.0
**Girls**																	
	once a day and less	57	29.2	234	29.1	255	30.2	236	26.7	257	28.2	151	26.2	17	40.5	1207	28.3
	twice a day	138	70.8	570	70.9	590	69.8	649	73.3	655	71.8	425	73.8	25	59.5	3052	71.7
	Total	195	100.0	804	100.0	845	100.0	885	100.0	912	100.0	576	100.0	42	100.0	4259	100.0

**Table 3 ijerph-16-02484-t003:** Proportion of response categories in frequency of toothbrush changing, visiting the dentist in last year and in gum bleeding stratified according to age (13–16 years of age) and gender.

	Group	Category	13 Years Old	14 Years Old	15 Years Old	16 Years Old	All Ages
			N (%)	N (%)	N (%)	N (%)	N (%)
**Changing toothbrushes**	whole sample	less often	397 (22.3)	382 (21.1)	279 (22.4)	20 (16.8)	1101 (21.7)
		at least once every 3 months	1385 (77.7)	1432 (78.9)	965 (77.6)	99 (83.2)	3966 (78.3)
		Total	1782 (100.0)	1814 (100.0)	1244 (100.0)	119 (100.0)	5067 (100.0)
	boys	less often	236 (25.6)	204 (22.4)	173 (25.9)	14 (18.2)	640 (24.3)
		at least once every 3 months	687 (74.4)	705 (77.6)	494 (74.1)	63 (81.8)	1993 (75.7)
		Total	923 (100.0)	909 (100.0)	667 (100.0)	77 (100.0)	2633 (100.0)
	girls	less often	161 (18.7)	178 (19.7)	106 (18.4)	6 (14.3)	461 (18.9)
		at least once every 3 months	698 (81.3)	727 (80.3)	471 (81.6)	36 (85.7)	1973 (81.1)
		Total	859 (100.0)	905 (100.0)	577 (100.0)	42 (100.0)	2434 (100.0)
**Preventive dental check-up**	whole sample	no	243 (13.7)	292 (16.1)	205 (16.5)	42 (35.6)	810 (16.0)
		yes	1534 (86.3)	1520 (83.9)	1040 (83.5)	76 (64.4)	4250 (84.0)
		Total	1777 (100.0)	1812 (100.0)	1245 (100.0)	118 (100.0)	5060 (100.0)
	boys	no	142 (15.5)	159 (17.5)	127 (19.0)	28 (36.4)	472 (17.9)
		yes	777 (84.5)	750 (82.5)	541 (81.0)	49 (63.6)	2158 (82.1)
		Total	919 (100.0)	909 (100.0)	668 (100.0)	77 (100.0)	2630 (100.0)
	girls	no	101 (11.8)	133 (14.7)	78 (13.5)	14 (34.1)	338 (13.9)
		yes	757 (88.2)	770 (85.3)	499 (86.5)	27 (65.9)	2092 (86.1)
		Total	858 (100.0)	903 (100.0)	577 (100.0)	41 (100.0)	2430 (100.0)
**Gum bleeding**	whole sample	never	842 (47.1)	834 (45.7)	514 (41.3)	44 (37.3)	2285 (44.9)
		occasionally + sometimes	831 (46.5)	876 (48.0)	645 (51.8)	54 (45.8)	2454 (48.3)
		always + often	116 (6.5)	115 (6.3)	86 (6.9)	20 (16.9)	347 (6.8)
		Total	1789 (100.0)	1825 (100.0)	1245 (100.0)	118 (100.0)	5086 (100.0)
	boys	never	475 (51.4)	458 (50.1)	298 (44.6)	33 (42.9)	1298 (49.2)
		occasionally + sometimes	394 (42.6)	398 (43.5)	322 (48.2)	33 (42.9)	1169 (44.3)
		always + often	55 (6.0)	58 (6.3)	48 (7.2)	11 (14.3)	173 (6.6)
		Total	924 (100.0)	914 (100.0)	668 (100.0)	77 (100.0)	2640 (100.0)
	girls	never	367 (42.4)	376 (41.3)	216 (37.4)	11 (26.8)	987 (40.4)
		occasionally + sometimes	437 (50.5)	478 (52.5)	323 (56.0)	21 (51.2)	1285 (52.5)
		always + often	61 (7.1)	57 (6.3)	38 (6.6)	9 (22.0)	174 (7.1)
		Total	865 (100.0)	911 (100.0)	577 (100.0)	41 (100.0)	2446 (100.0)

**Table 4 ijerph-16-02484-t004:** Crude effects (odd ratios (OR) and 95% confidence interval (95% C.I.)) of sociodemographic variables on oral health related behaviours and gum bleeding tested in binary logistic regression and tested individually.

		Tooth Brushing				Changing Toothbrushes			
		OR	(95% C.I.)	*p* value	N	OR	(95% C.I.)	*p* value	N
**Age** (continual)		0.96	(0.93 | 0.99)	0.012	8673	1.03	(0.95 | 1.11)	0.482	5074
**Gender** (reference: girls)		0.48	(0.44 | 0.52)	0.000	8673	0.73	(0.64 | 0.83)	0.000	5074
**Family affluence scale** (continual)		1.11	(1.09 | 1.13)	0.000	8617	1.03	(1.00 | 1.07)	0.033	3973
**Family affluence** (continual)		0.75	(0.71 | 0.80)	0.000	6849	0.78	(0.71 | 0.85)	0.000	4103
**Father’s education** (reference: University)				0.000	7002			0.042	4184
	Elementary	0.57	(0.44 | 0.73)	0.000	175	1.33	(0.88 | 2.00)	0.179	162
	Apprenticeship	0.60	(0.52 | 0.70)	0.000	995	0.98	(0.80 | 1.21)	0.870	923
	Secondary	0.90	(0.77 | 1.05)	0.174	951	1.30	(1.03 | 1.64)	0.027	754
**Mother’s education** (reference: University)				0.000	7016			0.014	4193
	Elementary	0.52	(0.40 | 0.69)	0.000	134	1.74	(1.04 | 2.90)	0.034	134
	Apprenticeship	0.59	(0.50 | 0.68)	0.000	683	0.80	(0.65 | 0.99)	0.043	646
	Secondary	0.76	(0.66 | 0.86)	0.000	1198	0.97	(0.79 | 1.18)	0.740	982
**Father’s employment** (reference: unemployed)		1.56	(1.29 | 1.88)	0.000	7118	0.78	(0.60 | 1.02)	0.937	4227
**Mother’s employment** (reference: unemployed)		1.43	(1.24 | 1.66)	0.000	7121	1.01	(0.81 | 1.26)	0.070	4232
		**Preventive Dental Check-Up**						**Gum Bleeding (Never)**	
		**OR**	**(95% C.I.)**	***p* value**	**N**	**OR**	**(95% C.I.)**	***p* value**	**N**
**Age** (continual)		0.87	(0.80 | 0.95)	0.002	5067	1.43	(1.28 | 1.60)	0.000	5086
**Gender** (reference: girls)		1.35	(1.16 | 1.57)	0.000	5067	0.88	(0.83 |0.94)	0.000	5086
**Family affluence scale** (continual)		1.2	(1.16 | 1.24)	0.000	3964	1.05	(1.02 | 1.07)	0.000	3978
**Family affluence** (continual)		0.88	(0.79 | 0.97)	0.011	4090	0.85	(0.79 | 0.91)	0.000	4108
**Father’s education** (reference: University)				0.000	4170			0.000	4188
	Elementary	0.18	(0.12 | 0.26)	0.000	119	0.59	(0.43 | 0.81)	0.001	74
	Apprenticeship	0.54	(0.41 | 0.70)	0.000	989	0.69	(0.59 | 0.83)	0.000	492
	Secondary	0.83	(0.61 | 1.12)	0.217	809	0.92	(0.76 | 1.10)	0.339	444
**Mother’s education** (reference: University)				0.000	4181			0.000	3485
	Elementary	0.13	(0.09 | 0.20)	0.000	85	0.48	(0.34 | 0.69)	0.000	52
	Apprenticeship	0.54	(0.41 | 0.70)	0.000	708	0.70	(0.59 | 0.83)	0.000	357
	Secondary	0.68	(0.53 | 0.88)	0.003	1063	0.78	(0.67 | 0.92)	0.003	559
**Father’s employment** (reference: unemployed)		2.23	(1.80 | 2.76)	0.000	4218	1.09	(0.87 | 1.38)	0.128	4234
**Mother’s employment** (reference: unemployed)		2.07	(1.58 | 2.70)	0.000	4221	1.15	(0.96 | 1.38)	0.448	4238

**Table 5 ijerph-16-02484-t005:** Mutually adjusted effects (odd ratios (OR) and 95% confidence interval (95% C.I.)) of sociodemographic variables on oral health related behaviours and gum bleeding tested in binary logistic and linear regression entered at once in one model.

		Tooth Brushing (10–16 Years) (N = 4333)			Changing Toothbrushes (13–16 Years) (N = 2843)		
		OR	(95% C.I.)	*p* value	OR	(95% C.I.)	*p* value
**Age** (continual)		1.05	(1.00 | 1.10)	0.066	1.04	(0.93 | 1.17)	0.502
**Gender** (reference: girls)		0.45	(0.39 |0.51)	0.000	0.75	(0.62 | 0.90)	0.002
**Family affluence scale** (continual)		1.07	(1.04 | 1.11)	0.000	1.01	(0.97 | 1.05)	0.726
**Family affluence** (continual)		0.80	(0.73 | 0.87)	0.000	0.78	(0.69 | 0.88)	0.000
**Father’s education** (reference: University)				0.007			0.022
	Elementary	1.05	(0.71 | 1.56)	0.796	1.43	(0.80 | 2.56)	0.230
	Apprenticeship	0.80	(0.66 | 0.96)	0.019	1.19	(0.91 | 1.55	0.203
	Secondary	1.08	(0.89 | 1.30)	0.433	1.52	(1.16 | 1.98)	0.003
**Mother’s education** (reference: University)				0.032			0.291
	Elementary	0.64	(0.42 | 0.99)	0.042	1.49	(0.74 | 3.00)	0.263
	Apprenticeship	0.77	(0.63 | 0.95)	0.013	0.84	(0.63 | 1.12)	0.234
	Secondary	0.84	(0.71 | 0.99)	0.038	0.91	(0.71 | 1.16)	0.448
**Father’s employment** (reference: unemployed)		1.28	(0.92 |1.78)	0.145	1.41	(0.92 | 2.17)	0.119
**Mother’s employment** (reference: unemployed)		1.33	(1.07 | 1.64)	0.009	0.99	(0.73 | 1.34)	0.942
		**Preventive Dental Check-Up** **(13–16 Years)** **(N = 2840)**			**Gum Bleeding (Never)** **(13–16 Years)** **(N = 2850)**		
		**OR**	**(95% C.I.)**	***p* value**	**OR**	**(95% C.I.)**	***p* value**
**Age** (continual)		0.98	(0.85 | 1.12)	0.739	0.90	(0.82 | 0.99)	0.033
**Gender** (reference: girls)		0.60	(0.48 | 0.75)	0.000	1.38	(1.19 | 1.61)	0.000
**Family affluence scale** (continual)		1.11	(1.05 | 1.17)	0.000	0.99	(0.96 | 1.02)	0.522
**Family affluence** (continual)		0.95	(0.82 | 1.10)	0.479	0.86	(0.78 | 0.96)	0.005
**Father’s education** (reference: University)				0.050			0.038
	Elementary	0.52	(0.30 | 0.92)	0.024	0.81	(0.51 | 1.27)	0.355
	Apprenticeship	0.77	(0.55 | 1.08)	0.136	0.82	(0.66 | 1.01)	0.065
	Secondary	1.03	(0.72 | 1.46)	0.889	1.08	(0.88 | 1.34)	0.454
**Mother’s education** (reference: University)				0.006			0.089
	Elementary	0.36	(0.20 | 0.65)	0.001	0.58	(0.35 | 0.96)	0.033
	Apprenticeship	0.90	(0.63 | 1.27)	0.547	0.81	(0.64 | 1.02)	0.075
	Secondary	0.85	(0.63 | 1.16)	0.310	0.85	(0.70 | 1.03)	0.099
**Father’s employment** (reference: unemployed)		1.74	(1.10 | 2.74)	0.017	1.05	(0.71 | 1.54)	0.811
**Mother’s employment** (reference: unemployed)		1.82	(1.34 | 2.47)	0.000	0.90	(0.70 | 1.15)	0.408

**Table 6 ijerph-16-02484-t006:** Mutually adjusted effects (odd ratios (OR) and 95% confidence interval (95% C.I.)) of sociodemographic variables on oral health related behaviours and gum bleeding tested in binary logistic and linear regression in model with eliminated insignificant predictors using forward conditional method. Presented are only models with variables with significant effect.

		Tooth Brushing (10–16 Years) (N = 4333)			Changing Toothbrushes (13–16 Years) (N = 2843)		
		OR	(95% C.I.)	*p* value	OR	(95% C.I.)	*p* value
**Age** (continual)		- - -	- - -	- - -	- - -	- - -	- - -
**Gender** (reference: girls)		0.45	(0.40 | 0.52)	0.000	0.75	(0.63 | 0.91)	0.003
**Family affluence scale** (continual)		1.08	(1.04 | 1.11)	0.000	- - -	- - -	- - -
**Family affluence** (continual)		0.81	(0.74 | 0.88)	0.000	0.76	(0.68 | 0.86)	0.000
**Father’s education** (reference: University)				0.011			0.013
	Elementary vs. University	1.04	(0.70 | 1.53)	0.853	1.56	(0.96 | 2.53)	0.074
	Apprenticeship vs. University	0.81	(0.67 | 0.98)	0.030	1.10	(0.88 | 1.38)	0.393
	Secondary vs. University	1.09	(0.90 | 1.31)	0.373	1.44	(1.13 | 1.85)	0.003
**Mother’s education** (reference: University)				0.044			- - -
	Elementary vs. University	0.65	(0.43 | 1.00)	0.049	- - -	- - -	- - -
	Apprenticeship vs. University	0.78	(0.64 | 0.96)	0.018	- - -	- - -	- - -
	Secondary vs. University	0.84	(0.71 | 1.00)	0.046	- - -	- - -	- - -
**Father’s employment** (reference: unemployed)		- - -	- - -	- - -	- - -	- - -	- - -
**Mother’s employment** (reference: unemployed)		1.33	(1.08 | 1.65)	0.008	- - -	- - -	- - -
		**Preventive Dental Check-Up ** **(13–16 Years)** **(N = 2840)**			**Gum Bleeding (Never) ** **(13–16 Years)** **(N = 2850)**		
		**OR**	**(95% C.I.)**	***p* value**	**OR**	**(95% C.I.)**	***p* value**
**Age** (continual)		- - -	- - -	- - -	0.90	(0.82 | 0.99)	0.024
**Gender** (reference: girls)		0.60	(0.48 | 0.75)	0.000	1.38	(1.19 | 1.61)	0.000
**Family affluence scale** (continual)		1.11	(1.06 | 1.17)	0.000	- - -	- - -	- - -
**Family affluence** (continual)		- - -	- - -	- - -	0.88	(0.80 | 0.96)	0.007
**Father’s education** (reference: University)				0.049			0.000
	Elementary vs. University	0.53	(0.30 | 0.92)	0.024	0.63	(0.44 | 0.91)	0.013
	Apprenticeship vs. University	0.76	(0.55 | 1.07)	0.117	0.74	(0.62 | 0.90)	0.002
	Secondary vs. University	1.02	(0.72 | 1.45)	0.918	1.01	(0.84 | 1.23)	0.889
**Mother’s education** (reference: University)				0.007			- - -
	Elementary vs. University	0.37	(0.21 | 0.66)	0.001	- - -	- - -	- - -
	Apprenticeship vs. University	0.90	(0.63 | 1.27)	0.539	- - -	- - -	- - -
	Secondary vs. University	0.85	(0.63 | 1.16)	0.302	- - -	- - -	- - -
**Father’s employment** (reference: unemployed)		- - -	(1.13 | 2.78)	0.013	- - -	- - -	- - -
**Mother’s employment** (reference: unemployed)		- - -	(1.34 | 2.48)	0.000	- - -	- - -	- - -

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
