# Peer review of "Self-Reported Oral Health Related Behaviour and Gum Bleeding of Adolescents in Slovakia in Relation to Socioeconomic Status of Their Parents: Cross-Sectional Study Based on Representative Data Collection"

_ijerph, 2019, doi:10.3390/ijerph16142484_

Round 1

Reviewer 1 Report

The paper shows that there were significant associations between parents’ education level/employment and self-reported oral health behaviors/symptom (gums bleeding) in a cross-sectional study among Slovakia adolescents.

This is an interesting study. However, there were some issues.

The paper should be revised.

ABSTRACT

1)    Please add the odds ratio with 95%CI.

2)    Please revise the conclusion because it’s not appropriate, and this is a cross-sectional study that showed temporal associations and did not investigate oral health status.

INTRODUCTION

1)    Please change the word, “But (L50)” to the word, “However”.

2)    Please add the hypothesis before the aim.

MATERIALS AND METHODS

1)    Please add the inclusion and exclusion criteria clearly.

2)    How did the authors treat the missing values? In the Table 1, the number was different. For example, in the Figure 1, the data among 10-12 years’ adolescents were shown in only brushing frequency. Please add the detail methods about the missing values in the text and also add each number in the Table 4-6. Furthermore, please add some comments about the final number of participants in the logistic analyses (the number of complete data).

3)    How did the authors investigate parents’ employment status and educational attainments? Please add the detail methods in the text.

4)    “U” of “Mann-Whiteny U-test” should be italic (L121).

RESULTS

1)    Please change the comma, “,” to period, “.” at each value (% or 95%CI) in all tables.

2)    Please avoid the duplicate, such as the Table 2 and Figure 1. Please delete or revise them.

3)    Please change the lines to bars in the Figure 1 because the data did not mean same persons. Furthermore, please add the statistical analyses and the results.

4)    Please add each number of participants in the Table 4. Which is a reference in each category? Please also clarify it; i.e., boys are reference or not. Furthermore, the targeted age group is different in each oral health behavior. Please revise it.

5)    Which is a reference in each category in the Table 5 and 6? Please revise the tables. Furthermore, please add the p values instead of “ns”.

DISCUSSION

1)    Please add the limitations in this study clearly; i.e., bias, a cross-sectional study, no data of oral health status, no data of important confounders, etc.

2)    All data were not collected among 10-16 years. Please revise the conclusion to avoid the misleading. Furthermore, revise the comments about “oral health” because the authors did not investigate the oral health status. Please delete the last sentence, “We recommended…inequalities”, because this is a cross-sectional study.

Author Response

Reviewer 1

Comment 1

The paper shows that there were significant associations between parents’ education level/employment and self-reported oral health behaviors/symptom (gums bleeding) in a cross-sectional study among Slovakia adolescents.

This is an interesting study. However, there were some issues.

The paper should be revised.

Reply 1: Thnank you for revision of our manuscript and for constructive comments that helped to impove our manuscript. Below we addressed all your coments and provided our reply and action that we made upon your comment. In general we understood that our term “oral health” was not used properly so we changed it throughout the text into “oral health related behaviours and gum bleeding” if referred to outcomes on our manuscript.

ABSTRACT

Comment 1.1: Please add the odds ratio with 95%CI.

Reply 1.1: Thank you for pointing out missing 95% CI values in abstract.

Adjustment in the text 1.1: 95% CI values were inserted into Results section of manuscript‘s Abstract

Results: We found that prevalence of OHRBs is slightly decreasing with age, and worse outcomes were reported by boys compared to girls (OHRB odds ratio range 0.45-0.75 (95%CI ranged 0.40-0.91), gum bleeding 1.38 (95%CI 1.19-1.61), p<0.05). OHRBs were in most cases significantly associated with socio-economic variables, lower affluence predicts worse outcomes (odds ratio range 0.76-0.88 (95%CI range 0.68-0.96), p<0.05).

Comment 1.2: Please revise the conclusion because it’s not appropriate, and this is a cross-sectional study that showed temporal associations and did not investigate oral health status.

Reply 1.2: Conclusion was revised and reformulated.

Adjustment in the text 1.2:

Our study provides representative findings on ORHBs in Slovakia and shows important associations of socioeconomic factors related with adolescents’ oral health issues.

INTRODUCTION

Comment 1.3: Please change the word, “But (L50)” to the word, “However”.

Reply 1.3: Thank you for suggestion.

Adjustment in the text 1.3: text was changed

Comment 1.4: Please add the hypothesis before the aim.

Reply 1.4: Hypothesis was added before the aim

Adjustment in the text 1.4: text was changed

It might be expected that better oral health related behaviours and prevalence of healthy gums would be associated with higher socio-economic status of the parents, such as higher education attainment, better family well-off.

MATERIALS AND METHODS

Comment 1.5: Please add the inclusion and exclusion criteria clearly.

Reply 1.5: We agree that exclusion and exclusion criteria might be mentioned in the text.

Adjustment in the text 1.5.: Requested information were added to the text of Sample and procedure paragraph.

In the second step, we obtained data from 8,902 adolescents from the fifth to ninth grades of elementary schools and prima to quarta of eight years grammar schools in Slovakia. Eligible respondents were all pupils in selected schools with age range from 10 to 17 years old (mean age 13.4; SD=1.4; 50.9 % boys). We excluded pupils from participating in the data collection who provided parent’s disagreement with participation in the HBSC study, pupils who were not present in the school and pupils who individually refused to participate. For the purposes of the study we excluded also 17 years old who consisted small group (N=8).

Comment 1.6: How did the authors treat the missing values? In the Table 1, the number was different. For example, in the Figure 1, the data among 10-12 years’ adolescents were shown in only brushing frequency. Please add the detail methods about the missing values in the text and also add each number in the Table 4-6. Furthermore, please add some comments about the final number of participants in the logistic analyses (the number of complete data).

Reply 1.6: Missing values were treated listwise, it means, that we did not exclude participants from the data set if they have some variable missing. They were excluded just from analysis on particular association as is usual in similar studies.

Adjustment in the text 1.6. Notion about pairwise removal of the participants with missing values.

As item non-response occurred in non-systematic way, our total responses on particular tested variables varied. For data analysis we used list-wise case deletion regarding each analysis.

Comment 1.7: How did the authors investigate parents’ employment status and educational attainments? Please add the detail methods in the text.

Reply 1.7: items investigating parents’ SES were added to the Measures section.

Adjustment in the text 1.7:

We collected data on sociodemographic characteristics (age, gender), socioeconomic indicators (family affluence scale (FAS, score from 0-13, higher score indicates more affluence) [28], self-rated family well off (How well off do you think your family is?, answer categories: 1-very well off” to 5“not well off at all”), parents’ employment status (Does your father/mother have a job?, answer categories: 1-yes, 0-no) and education attainments (What school, what kind of education has your father/mother finished?, answer categories: 1-elementary, 2-apprenticeship, 3-secondary, 4-university) and data on oral health…

Comment 1.8:“U” of “Mann-Whiteny U-test” should be italic (L121).

Reply 1.8: thank you for the recommendation

Adjustment in the text 1.8.: U-test was changed for italic “U-test”

RESULTS

Comment 1.9: Please change the comma, “,” to period, “.” at each value (% or 95%CI) in all tables.

Reply 1.9: You are right, we changed that

Adjustment in the text 1.9.: Number with decimal commas was changed into numbers with decimal period throughout all the text.

Comment 1.10: Please avoid the duplicate, such as the Table 2 and Figure 1. Please delete or revise them.

Reply 1.10: You are right, figures, duplicated the results from table 2.

Adjustment in the text 1.10.: Figure was removed

Comment 1.11: Please change the lines to bars in the Figure 1 because the data did not mean same persons. Furthermore, please add the statistical analyses and the results.

Reply 1.11: Figure was removed because it duplicated the results from table 2

Adjustment in the text 1.11.: Figure was removed

Comment 1.12: Please add each number of participants in the Table 4. Which is a reference in each category? Please also clarify it; i.e., boys are reference or not. Furthermore, the targeted age group is different in each oral health behaviour. Please revise it.

Reply 1.12: The N was added to each analysis, and age group is different only regarding tooth brushing where this item was used in all age groups and the other outcomes were administered in participants aged 13years and more. Girls were reference category…

Adjustment in the text 1.12.: The N was added to each table correspondingly to each analysis. Reference categories were stated clearer.

Comment 1.13: Which is a reference in each category in the Table 5 and 6? Please revise the tables. Furthermore, please add the p values instead of “ns”.

Reply 1.13: Reference category was stated clearer. And p values were added

Adjustment in the text 1.13.: Reference category was stated explicitly. And p values were added to the each coefficient.

DISCUSSION

Comment 1.14: Please add the limitations in this study clearly; i.e., bias, a cross-sectional study, no data of oral health status, no data of important confounders, etc.

Reply 1.14: New paragraph with strengths and limitations of the study was added

Adjustment in the text 1.14.: New paragraph with strengths and limitations of the study was added

4.1 Strenghts and limitiations

As a strength of our study we consider the fact that this is the first study assessing the association of socioeconomic factors on self-reported oral health related behavior and gum bleeding among Slovak adolescents. Another strength is large and nationally representative sample based on internationally elaborated study protocol.

The limitations of the present study should be noted as well. First, the cross-sectional design allows to make inferences only about associations and not about causality. Second, self-reported nature of the data did not allow us to get actual picture of the children oral health status which might be possible only using dentist examination, what was not feasible within the study design. However, items used for monitoring the epidemiological situation in Slovakia are specifically selected to cover main issues dealt with in oral health promotion and problem prevention. Third, we did not studied effect of potential confounders that might affect the presented associations to a certain degree. However, we aimed to provide clear message on socio-economic factors associations and the relevance of those factors for individual outcomes. Assessing various confounders was not the aim of our study. Fourth study limitation is relatively high number of missing responses to individual items covered by this study what in combination in regression models cumulates and lower sample size. Regarding the nature of the study we consider those nonresponses as random and not affecting the associations reported here but lowering the statistical power of the sample. Last limitation is using different length of the questionnaire in three age groups (10-12, 13-14, 15 and more) because of different ability to fill in the questionnaire in those age groups. The youngest respondents had shortest questionnaire and thus items on toothbrush changing, preventive dental check-up and gum bleeding were not included in their questionnaire. Within such complex study such HBSC is not always feasible to cover all topic in all age categories and some selection had to be made in the favor of effectivity and feasibility.

Comment 1.15: All data were not collected among 10-16 years. Please revise the conclusion to avoid the misleading. Furthermore, revise the comments about “oral health” because the authors did not investigate the oral health status. Please delete the last sentence, “We recommended…inequalities”, because this is a cross-sectional study.

Reply 1.15:

Adjustment in the text 1.15.: We made clearer age distinctions between different outcomes and deleted the last sentence that overestimated the impact of cross-sectional study designs.

Our study provides representative findings on oral health related behaviour and gum bleeding in Slovakia, showing that prevalences of tooth brushing, dental check-up are under the average of the European countries and with relatively high proportion of gum bleeding among adolescents aged 10-16 regarding tooth brushing and 13-16 years of age regarding dental check-up, gum bleeding. We found that girls tended to reported better oral health outcomes comparing to boys and that younger respondents tended to report better oral health outcomes compared to older ones. The study also supports importance of socioeconomic factors and their association with oral health outcomes indicating, that lower socioeconomic situation of the adolescents’ family, the worse is adolescents’ health.

Reviewer 2 Report

This study addresses the relationship between several socio-demographic indicators, on the one hand, and the selected indicators of oral health behavior (dental hygiene, 21 toothbrush changing, preventive check-up) and general oral health (gums bleeding), on the other. The sample has been drawn from the Health Behavior in School-aged Children study, a representative, national school-based survey of adolescents from Slovakia. The survey is based on weighted random sample of 140 schools and encompasses 8,902 adolescents, with age range from 10 to 17 years old. Socio-demographic variables of interest included age, gender, family affluence, self-rated family SES, parents’ employment status and education attainment. Binary logistic regression was used to estimate the effects of socio-demographic variables on oral health variables.

The novelty of this study lies in the fact that the study was conducted in Slovakia, a country about which little is known from the medical, and, even less from the epidemiological point of view. However, the relationship between gender, age and socio-economic status, on the one hand, and oral health, on the other, has extensively been studied. Therefore, the results are hardly new. I encourage the authors to emphasize how their study is different from the abundant empirical literature on the topic and how this work advances our understanding of social predictors of oral health among adolescents.

Author Response

Reviewer 2

Comment 2

This study addresses the relationship between several socio-demographic indicators, on the one hand, and the selected indicators of oral health behavior (dental hygiene, 21 toothbrush changing, preventive check-up) and general oral health (gums bleeding), on the other. The sample has been drawn from the Health Behavior in School-aged Children study, a representative, national school-based survey of adolescents from Slovakia. The survey is based on weighted random sample of 140 schools and encompasses 8,902 adolescents, with age range from 10 to 17 years old. Socio-demographic variables of interest included age, gender, family affluence, self-rated family SES, parents’ employment status and education attainment. Binary logistic regression was used to estimate the effects of socio-demographic variables on oral health variables.

The novelty of this study lies in the fact that the study was conducted in Slovakia, a country about which little is known from the medical, and, even less from the epidemiological point of view. However, the relationship between gender, age and socio-economic status, on the one hand, and oral health, on the other, has extensively been studied. Therefore, the results are hardly new.

Comment 2.1

I encourage the authors to emphasize how their study is different from the abundant empirical literature on the topic and how this work advances our understanding of social predictors of oral health among adolescents.

Reply 2.1:

We agree that there is a large body of research related to oral health and socioeconomic factors. Most of that research was conducted on adult samples or adolescent above 15 years of age. Epidemiological data on younger adolescents and from nationally representative samples are not so frequent. In recent years most of research about oral health comes from developing countries and research from developed countries is focused on specific oral health issues. Epidemiological data that provides broader overview about the oral health related behaviour among adolescents, which determines there oral health in the adulthood, is not so common, and in Slovakia and neighbouring countries such data are not available (to our knowledge).  Regarding association of oral health behaviours and socioeconomic status, the association was established and confirmed in many countries, but it is still good to provide evidence and support such association even on new samples and countries where such association may be reasonably expected but nobody confirmed it using nationally representative samples. It also might provoke further study of adolescents oral health related factors in Slovakia and neighbouring countries.

Adjustment in the text 2.1

We formulated new paragraph on Strengths and limitation (in Discussion) where we mentioned, that our study brought data that were lacking in the dentist community.

4.1 Strengths and limitations

As a strength of our study we consider the fact that this is the first study assessing the association of socioeconomic factors on self-reported oral health related behavior and gum bleeding among Slovak adolescents that were lacking for a long time in the dentist community. Another strength is large and nationally representative sample based on internationally elaborated study protocol.

Round 2

Reviewer 1 Report

The paper was overall improved. However, there was some issues. Please revise them.

1) Please add each number of participants and revise the tables because nobody knows the results of cross tabulation; i.e., how many fathers at each education level did brush their teeth once per day?

2) How did the authors calculate the crude odds ratios in continuous values? It’s impossible.

Author Response

Comments and Suggestions for Authors

The paper was overall improved. However, there was some issues. Please revise them.

Reply:

Thank you for appreciating the adjustment of the manuscript and its improvement.

1) Please add each number of participants and revise the tables because nobody knows the results of cross tabulation; i.e., how many fathers at each education level did brush their teeth once per day?

Reply 1:

Thank you for your comment. We tried to implement it but we think that there might be some misunderstanding. First we would like to make clear, that our paper was focused on oral health related behaviour of adolescents and not their parents, so we are not able to provide any data or number about fathers and their frequency of tooth brushing. Probably you wanted to add numbers of the respondents reporting different level of education of their parents related to respondent’s oral health related behaviour.

Regarding comments from previous revision we have added numbers of participants entering each level of logistic regression, due to nonresponses their numbers differed in each tested variable. Adding additional number of participants in each category (of parents educational attainment) would be necessary if we would be using cross tabulation and chi-squared tests. Because we used logistic regression, and differences between variables are expressed as odds ratios, the number of participants in each category is redundant and not usual. This is really the first time that we are asked to provide such information in that kind of analysis. However, we provided those numbers in Table 4

Adjustment in the text 1

We added numbers of respondents in category of father’s and mother’s educational attainment (elementary, apprenticeship and secondary) in Table 4.

2) How did the authors calculate the crude odds ratios in continuous values? It’s impossible.

Reply 2:

Odds ratio is numerical expression of the strength of the association between the variables. Crude odds ratio means, that we assessed only association of two variable – predicting variable (or independent variable) and outcome (dependent variable), without any adjustments. However logistic regression allows to generalize the odds ratio beyond two binary variables and allow to use as a predictor even a continuous variable. Using logistic regression to calculate odds ratio for continuous variables is not just possible, but pretty common. In binary logistic regression only outcome variable has to be categorical with two values (usually 0 vs 1), but as predictors categorical and continuous variables might be used in various compositions and models (adjustment, interaction etc.).

Odds ratio (e.g. OR=2.6) of the outcome predicted by the continuous variable (age, FAS score, etc.) might be interpreted as one unit increase in continuous variable leads to a e.g 2.6 fold increase in the odds of the outcome.